Data-driven flight path monitoring technique using recurrent neural network for the safety management of commercial aircraft

Kim Naeun 1 2 hkim439@asu.edu
http://orcid.org/0000-0001-6823-1946 Hamza Mohamed H. 2
Koh Bong-Hwan 3
1 Department of Mechanical Engineering, Dongguk University , Seoul , South Korea
2 School for Engineering of Matter, Transport, and Energy, Arizona State University , Tempe, AZ , United States
3 Department of Mechanical, Robotics and Energy Engineering, Dongguk University , Seoul , South Korea
Pires Ivan Miguel
Electronic publication date: 2025 Mar 17
Publication date: 2025
Volume: 11
Electronic Location ID: e2753
Received 2024 Mar 8; Accepted 2025 Feb 19
Copyright: © 2025 Kim et al.
Copyright year: 2025
Copyright holder: Kim et al.
License: This is an open access article distributed under the terms of the Creative Commons Attribution License, which permits unrestricted use, distribution, reproduction and adaptation in any medium and for any purpose provided that it is properly attributed. For attribution, the original author(s), title, publication source (PeerJ Computer Science) and either DOI or URL of the article must be cited.
License URL: https://creativecommons.org/licenses/by/4.0/

Keywords: Incipient spin detection, Flight safety, Machine learning-based flight monitoring, Air traffic management

Funding: Dongguk University-Seoul Research Fund 2023 This work was funded by the Dongguk University-Seoul Research Fund of 2023. The funders had a role in the preparation of the manuscript and decision to publish. The funders had no role in study design, data collection and analysis.

==============================
Aviation spins, particularly at low altitudes, significantly contribute to fatalities due to limited recovery time. Standard recovery procedures typically only become eligible after a spin is fully developed, by which time multiple turns may have already resulted in substantial altitude loss. The primary challenge in upset prevention is heavy reliance on the pilot’s situational awareness, which is only effective before the spin has been fully developed. To address this issue, this study proposes an early detection capability to significantly enhance immediate response actions, potentially mitigating altitude loss and enabling pilots to recognize the initial signs of upset conditions. This research introduces a real-time predictive tool based on a novel recurrent neural network (RNN) model that utilizes data from the NASA Generic Transport Model (GTM)-a research platform designed for experimental flight case studies-to predict nonlinear flight responses during the critical initial seconds of a spin. Rigorous validation against ground truth data demonstrates the RNN model’s superior predictive capabilities in detecting incipient spin phase, offering an essential tool for proactive spin management and reducing the risk of ground collisions. This early detection capability empowers pilots to identify the initial signs of upset conditions and make informed operational decisions, ultimately improving aviation safety. This advancement underscores the potential of advanced machine learning technologies to transform safety protocols by enabling earlier and more effective intervention strategies, thereby preempting catastrophic events.

Introduction

Since the early years of flight development, all aircraft have encountered aerodynamic stall states, often leading to severe consequences. Aerodynamic stall upset accidents involving commercial aircraft account for more fatal and serious injuries than any other type of accident (Anderson, 1979; Lee et al., 2020). According to the Aircraft Owners and Pilots Association (AOPA) Air Safety Institute, stall upset accidents have a fatality rate of approximately 28%, higher than the rates observed in other types of upset scenarios (Landsberg, Hummel & Murphy, 2003). One of the most critical stall upset conditions is the aircraft spin, a hazardous scenario characterized by rapid descent rates and the significant altitude required for recovery (Bennett & Lawson, 2018). Flight spin upsets are among the most hazardous scenarios in commercial aviation, accounting for 65–70% of fatalities despite representing only about 7% of total reported flight accidents (Steele, 2018). These events are particularly catastrophic because they often occur at low altitudes, where rapid descent rates and the significant altitude required for recovery maneuvers leave pilots with minimal opportunity to respond effectively (Bennett & Lawson, 2018). Research on flight spin upsets has been extensive due to their complex aerodynamics and critical recovery challenges. Historical efforts, such as the one by Khrabrov, Sidoryuk & Goman (2013) has delineated the aerodynamic triggers and phases of spin development, providing foundational insights into conditions that increase an aircraft’s propensity for hazardous operational states. Furthermore, traditional recovery techniques, such as those recommended by the U.S. Federal Aviation Administration, emphasize recovery after a spin is fully developed (Federal Aviation Administration (FAA), 2022). However, these methods are often insufficient under low-altitude conditions, as they typically initiate too late to counteract the rapid descent rates involved (Landsberg, Hummel & Murphy, 2003). Applying these insights to practical scenarios, where environmental variability and pilot responses play significant roles, remains challenging. This research aims to bridge this gap by focusing on the early detection and real-time management of spin upsets, an area where existing models continue to fall short.

This study proposes an aerodynamic upset prognosis framework based on a recurrent neural network (RNN) architecture. The scenario of spin upset is investigated using a machine learning (ML) model designed to replicate critical conditions and provide real-time flight path trajectory predictions. This framework is crucial for diagnosing aircraft anomalies and extends to the generation of a statistically informed flight path envelope, contributing to enhanced aviation safety management (Hamza et al., 2022). The reference (ground truth) model is NASA’s Generic Transport Model (GTM), which has been utilized for training data acquisition using MATLAB/Simulink. Stochasticity in the initial upset conditions was introduced to create a ground truth repository of aircraft spin trajectories. The RNN-based model is subsequently trained to minimize the discrepancy between the ground truth data and the ML-predicted trajectories.

This article is organized as follows: “Related work” reviews historical efforts to address flight spin phenomena, their aerodynamic characteristics and recovery methods towards flight development and describes the development of simulations and deep learning (DL)/ML models to mitigate flight upset conditions. Next, “Methodology” introduces both the RNN-based prediction model and the reference model, GTM, and the acquisition of training data. “Results and Discussion” presents the prediction results, validation of the proposed model, optimization details, and an evaluation of prediction accuracy against ground truth data. Finally, “Conclusions” addresses the contribution of this study towards the development of aviation safety and limitations with potential avenues for future research.

Related work

An overview of recent developments in spin prediction, recovery methods, and ML integration in flight disturbance management is provided in this section. It begins with a broad focus on understanding the aerodynamic factors that contribute to spin scenarios, then examines historical efforts in spin recovery techniques and the role of ML in improving spin prediction and management. The content is organized into three subsections: Spin Prediction and Early Detection, Spin Recovery Techniques, and ML for Flight Upset Detection and Recovery.

Spin prediction and early detection

This subsection reviews the literature on spin prediction and early detection, emphasizing the challenges of translating theoretical insights into practical applications. Traditionally, aerodynamic models for analytical spin studies have been developed using experimental data obtained from a variety of dynamic wind tunnel tests. These tests include rotary balance coning motion tests, oscillatory coning motion tests, and large-amplitude forced oscillation tests, all conducted across a wide range of aerodynamic angles and control surface deflections (Murch & Foster, 2007). An alternative method, referred to as system identification, involves developing an aircraft spin model by deriving aerodynamic forces and moments directly from the flight data of a spinning aircraft, as proposed by Morelli & Klein (2016). A three-degree-of-freedom spin model was developed to predict spin phases for a fighter configuration that experiences yawing moment asymmetry at high angles of attack (Malik, Akhtar & Masood, 2017). Further studies in the literature provided crucial insights into the stall and post-stall aerodynamics that led to spin conditions, identifying specific flight conditions that predispose aircraft to dangerous states (Chambers, 1980; Khrabrov, Sidoryuk & Goman, 2013; Marcinkiewicz, Goraj & Figat, 2019).

While existing research primarily focuses on post-spin recovery strategies, this study aims to address a critical gap in aviation safety by focusing on the early detection and prediction of incipient spin phase (specifically within the first 0–4 or 6 s). This shift in focus from reaction to prediction is crucial because the rapid progression of spin at low altitudes necessitates immediate intervention to avert catastrophic outcomes. Accurately predicting a spin within this critical timeframe allows pilots to be alerted in real time, enabling timely corrective actions or triggering automated recovery systems, thereby preventing a full-blown spin and significantly enhancing flight safety. This proactive approach aligns with U.S. FAA initiatives and provides a novel framework for significantly improving the effectiveness of safety protocols by equipping pilots with the critical tools needed for immediate and effective spin management.

Spin recovery techniques

Building on the advancements in spin prediction and early detection discussed in the previous subsection, this subsection examines spin recovery techniques. Various approaches have been developed, demonstrating significant potential for enhancing spin management effectiveness. Techniques such as momentum vector control have shown promise in recovering from both steep and flat spins (Lee & Nagati, 2004). Bifurcation analysis, as an application of dynamical system theory, has been utilized to investigate various nonlinear flight dynamics problems. It allows the identification of different spin characteristics, including steep or flat, steady or oscillatory, right or left, and inverted spins. Additionally, it can reveal transitions from low-angle-of-attack trimmed flight to steady or oscillatory spin states, shifts between spin states, and the onset of wing rock phenomena. Bifurcation analysis provides valuable insights for developing piloting strategies for spin recovery and designing flight control laws aimed at automatic spin recovery and spin prevention (Malik, Masud & Akhtar, 2020). Rao & Go (2019) developed a spin recovery algorithm by solving a trajectory optimization problem using the multiple shooting method, where time and altitude loss were incorporated into the cost function to be optimized. Furthermore, the NASA standard spin recovery protocol, which emphasizes counteracting spin rotation, has gained wide acceptance within the industry (Chambers, Bowman & Malcolm, 1975; Chambers & Stough, 1986; Stowell, 2007; Bunge & Kroo, 2018). Traditional recovery techniques remain valuable but often prove insufficient in low-altitude scenarios where rapid response is critical. Automatic spin recovery control systems are less commonly employed compared to spin prevention systems (Buckner, Walker & Clark, 1979; Bennett & Lawson, 2018) because fully developed spins, particularly oscillatory flat spins, typically occur at very high angles of attack. In these conditions, aerodynamic control surfaces become relatively ineffective and are often inadequate for successful spin recovery actions. To address these limitations, there is a growing emphasis on integrating recovery techniques with proactive, predictive models. This integration is crucial for developing a more responsive spin management system capable of anticipating and mitigating potential spins before they fully develop.

Machine learning for flight upset detection and recovery

Recent advancements in aircraft trajectory modeling have increasingly incorporated ML algorithms capable of handling complex, multi-dimensional data in real time (Ayhan & Samet, 2016; Lymperopoulos, Lygeros & Lecchini, 2012; Pang & Liu, 2020), significantly enhancing the accuracy and efficiency of flight upset detection and recovery systems. With its ability to process large volumes of sensor data and recognize patterns in real time, ML addresses the limitations of traditional recovery methods discussed in the previous subsection by enabling adaptive control strategies (Lu et al., 2023). The integration of ML into the early detection and intervention process, particularly in the context of spin dynamics, represents a pivotal shift in flight safety protocols. This evolution is supported by the Next Generation National Air Transportation System initiative, led by the U.S. FAA, which emphasizes the need for adaptive, data-driven solutions capable of responding to the limited time available during critical in-flight emergencies (Darr, Ricks & Lemos, 2008).

Several studies in the literature have focused on predicting flight trajectories and quantifying uncertainties under stall conditions. Research by Zhang & Mahadevan (2020) and Babl & Engelbrecht (2020) have demonstrated the potential of Bayesian neural networks (BNNs) and dynamic models in predicting critical flight trajectories and managing deep stall scenarios effectively. These studies emphasize the growing recognition of ML as a transformative tool for addressing flight upset conditions. BNNs, in particular, excel in quantifying uncertainty in predictions, making them well-suited for safety-critical applications like flight trajectory prediction, where real-time decision-making is essential (Pang & Liu, 2020). For example, probabilistic aircraft trajectory prediction considering weather uncertainties has been explored using dropout as a Bayesian approximate variational inference method (Feng et al., 2023; Pang & Liu, 2020). While BNNs effectively quantify uncertainty, they may fall short in modeling time-dependent dynamics. Models such as Bayesian RNNs and deep residual RNNs (DRR-RNNs) address this limitation by capturing temporal dependencies, offering enhanced predictions for evolving aerodynamic scenarios. The adoption of these advanced models, as investigated by Pang et al. (2021) and Yu, Yao & Liu (2019), represents a significant technological leap, providing more accurate predictions and improved management of aircraft aerodynamic responses. By leveraging integral data, these models establish a proactive framework capable of predicting and mitigating risks before they escalate into serious upsets. A robust real-time aircraft trajectory prediction model was developed to address high-altitude stall upset scenarios using long short-term memory (LSTM) neural networks (Hamza et al., 2023). In this work, the trajectory prediction was computed orders of magnitude faster than a high-fidelity aerodynamics simulator. This research builds on the capabilities of current ML models by specifically focusing on spin detection and real-time management, areas that are not sufficiently addressed by existing approaches.

Reinforcement learning (RL)-based optimal controller was developed to recover unmanned aerial vehicle (UAV) from stable spin mode to low alpha flight regime in relatively short time (Kim et al., 2017). Zhu et al. (2019) developed a deep RL (DRL)-based model-free flat spin recovery scheme to recover miniature UAVs back to steady level flight swiftly. A UAV upset recovery system was developed using offline RL to discover and improve recovery strategies, which are then integrated into an online component for real-time decision-making during upsets (Dutoi et al., 2008). The system architecture is divided into two components: one for high angular rate upsets and another for unusual attitude upsets. By designing compact input and output sets, the system ensures RL applicability while reducing complexity. NASA’s GTM is used for training, evaluation, and robustness testing. Results demonstrate that the learning process often improves known recovery strategies and that the learned strategies are robust to uncertainty. These ML-based spin recovery strategies are effective as long as the incipient spin is detected early enough. Therefore, our proposed RNN-based trajectory prediction and spin detection framework can be integrated with spin recovery techniques to enhance their effectiveness, enabling a fully automated spin detection and recovery system.

Methodology

In this study, high-fidelity flight dynamics simulations are conducted to generate a stochastic repository of ground truth data under spin upset conditions. The design configuration of the NN model, selection of hyperparameters, and optimization methods suitable for predicting nonlinear problems and ensuring stable ML training are thorough discussed. Thereafter, the GTM framework and the acquisition of ground truth data during the main simulation phases of flight spin upset and the detailed aerodynamics involved are discussed. Figure 1 shows a summary of the overall framework. Ground truth data is generated using MATLAB/Simulink GTM simulations, thereafter series of data preprocessing are performed which are fed into the NN model for training and validation. Various metrics for results evaluations and accuracy assessment are conducted.

Figure 1 Schematic of the proposed model framework for spin trajectory prediction and aviation safety.

NN training methodology

Traditional NNs showed successful results in solving multidimensional classification and regression problems. Recently, models are being developed to solve various complex problems using deep NNs (DNNs) with complex architecture and layers (Gal & Ghahramani, 2016; Feng & Huang, 2021; Shao, Si & Zhang, 2022; Zhang et al., 2023; Chenglong Bao et al., 2023). With vast advancement in technology, traditional NNs fail to tackle cutting-edge problems with complex nature. ML has specialized NNs, which can be used for specific tasks, such as Convolution neural networks (CNNs) for image processing and analysis, and RNNs/LSTMs for data series predictions.

An RNN-based model provides modular designs that can be trained on different temporal input data, such as weather forecast, stock market, and the mechanics of materials modeling data. Furthermore, the main advantage of the RNN structure is that information about the previous state can be stored in the form of history variable (memory), as shown in Fig. 2, which becomes powerful to deal with time-series data by providing the temporal dependencies (Hopfield, 1982). Equations (1) and (2) summarize the mathematical formulation of the RNN, as follow:

Figure 2 Schematic of the RNN cell; where xt is the model input variable, ht is the history variable of the RNN at time t, and A is the RNN activation function.

(1) ht=tanh⁡(Whhht−1+Wxhxt+bh)

(2) yt=Whyht

where, ht is the history variable, yt is the predicted trajectory vector, xt is the input parameter vector at time t, Wxh, Whh and Why are the learnable weight matrices of the RNN that link the input parameters to the predicted trajectory, for example Wxh links xt to ht, and likewise for the other matrices. The main spin controlling parameters, as discussed in the next subsection, are used as the RNN-based model input, where the parameters initial conditions and their evolution with time obtained through GTM simulations are passed as the model input matrix, while the altitude as a function of time is selected as the model output vector.

RNN can capture the temporal dependency of the spin trajectory data due to the presence of history variable. However, due to such temporal dependencies, optimization of the NN will include backpropagation of the cost function through layers and time. Consequently, RNN will suffer from vanishing gradients, where the weight matrices will not get updated during training, especially in highly nonlinear problems, such as trajectory prediction under an aerodynamic upset. Hence, this is considered as one of the main limitations of the current model. By adding extra hidden layers with batch normalization, the prediction accuracy and training stability can be improved.

Traditional DNN operates in a feedforward manner, thus treating each time step prediction independently and cannot correlate the state parameters of a given problem at the current and previous time steps (Borkowski, Sorini & Chattopadhyay, 2022). RNN architectures and their variations incorporate such dependencies by having a hidden state “internal memory” and hence are suitable for learning the governing laws of time series data. Given a training dataset, RNN architectures result in more accurate training and validation accuracies at enhanced convergence rate compared to feedforward DNN (Gulli & Pal, 2017). Aircraft trajectory prediction applications have temporal flow, from take-off to landing, which has the distinctive characteristics of time series data. Therefore, the RNN-based model is considered in this study for real-time aircraft spin prognosis and subsequent trajectory prediction, as shown in Fig. 3. Coupling hidden layers with batch normalization on the top of RNN layers can affect the training accuracy and speed (Gulli & Pal, 2017). The selected ML configuration consists of two RNN layers, followed by five hidden layers with batch normalization applied between layers, and finally a fully connected output layer. The choice of the number of RNN and hidden layers is based on model tunning, where parametric study at the early model development phase has been conducted to build a NN architecture with minimal possible number of learnable parameters, while conserving prediction accuracy. Batch normalization is applied to maintain NN layer output distribution, which consequently improves training convergence. To mitigate overfitting and optimize the NN model, the GTM data is split into 80% for training and 20% for validation. A k-fold Cross-Validation with k = 5 is then conducted. The selection of model attributes, including the loss function and learning method, significantly influences the performance of the surrogate model. The root mean square error (RMSE) loss function (Amos & Kolter, 2017), computed by comparing GTM data with model predictions, is minimized using the gradient descent method. This optimization process determines the learnable parameters of NN, such as weights and biases. The Adam optimizer, an adaptive stochastic gradient descent method with a learning rate of 2×10−5, facilitates this process. The first and second moment hyperparameters are set to 0.9 and 0.999, respectively. Backpropagation, employing the chain rule, updates these learnable parameters across all NN layers during each training step.

Figure 3 Schematic of the proposed RNN-based model framework; input variables of model are the equivalent airspeed, angle of attack, sideslip, angular velocities, and the output variable is the simulation trajectory.

The input parameters and output trajectory for both configurations are processed through standardization scaling, to transform the data distribution to a standard normal distribution. This allows the NN to converge faster, since all the input parameters will be of the same units, while as well, the loss function will experience less numerical approximation and machine errors. Finally, dropout is applied for the hidden layers to avoid overfitting as discussed in the Results and Discussion section. The hardware setup for both RNN training and testing, as well as GTM simulations is AMD Ryzen 7 3700X (eight cores with a 3.6 GHz core clock), NVIDIA TITAN RTX graphics processing unit (GPU) (4,608 CUDA cores with a 1.365 GHz core clock) and with 64 GB DDR4 memory running on a Windows 10 (64-bit) operating system.

Generic transport model with flight spin upset

The GTM is a high-fidelity flight dynamic simulation model implemented into MATLAB/Simulink. It is a 5.5% dynamically scaled model developed for a twin-jet commercial transport aircraft established within the Integrated Resilient Aircraft Control (IRAC) project as shown in Table 1 (Jordan et al., 2004), which analyzes critical flight conditions for the aviation control system (Hueschen, 2011). The model invokes the nonlinear six-degree-of-freedom aerodynamic relations, as well as wind tunnel data, thrust models, control system, and geometry and mass properties (Bacon & Gregory, 2007). The GTM capability includes aerodynamics modeling, experimental flight testing, and flight trajectory generation parameters for advanced research regarding upset flight conditions (Jordan et al., 2004). Figure 4 summarizes the overall framework of GTM. Table 1 shows a comparison of the parameters of the full scale and the subscale aircraft model.

Table 1 Comparison of the parameters of full scale and the subscale aircraft model (Jordan et al., 2004).

	Length	Wingspan	Weight	
Full scale aircraft	145.5 ft	124 ft	200,000 lb	
5.5% Model	96 in	82 in	49.6 lb	
	Roll inertia	Airspeed	Altitude	
Full scale aircraft	2.64×106sl−ft2	320 mph	13,000 ft	
5.5% Model	1.33sl−ft2	75 mph	1,000 ft	

Figure 4 Schematic of the GTM framework, which has subsystems labeled thrust, engine, gravity, wind tunnel that contain computation of the aerodynamics with six-degree-of-freedom state variables as input for the full state of the aircraft at time increment ∆t.

GTM provides a high-fidelity tool for evaluating the controlling parameters of the aircraft at different flight phases. Thus, this high-fidelity simulator can be used for flight path investigation and planning purposes; however, it cannot be implemented for real-time aircraft dynamic response monitoring, due to the computational loading emerging from numerically solving nonlinear differential equations at each solution time step (Jordan et al., 2004). To avoid such limitation, the proposed RNN-based model can establish an online platform for flight path anomaly detection, as well as subsequent spin recovery optimization, using the high-fidelity control surface deflections and trajectories provided by GTM simulations as training data.

GTM is used to simulate 2,500 random spin trajectories each with a total time of 20 s. Upon convergence study, 5×10−3 s is used as a timestep (∆t) to ensure stability. The main GTM aerodynamic target conditions controlling the aircraft spin behavior are selected to be the equivalent airspeed (eas) and the true airspeed (tas), which are calibrated airspeeds corrected for compressibility, and temperature and pressure, respectively, angle of attack (α), side slip (β), flight path angle (γ), and angular velocities (p, q, and r). The target conditions were randomly distributed to generate different trajectories with mean and standard deviation, as summarized in Table 2. The mean of the target conditions was carefully selected to initiate a spin condition in the GTM simulations upon sensitivity analysis study. This stochasticity generates various trajectories accounting for the spin uncertainties, which is consequently used for reliable ML training.

Table 2 GTM target conditions with mean, and standard deviation.

Modified target conditions	Mean	Standard deviation	
eas (knots)	70.66	7.066	
tas (knots)	76.497	7.6497	
α (deg)	21.34	2.134	
β (deg)	−1.33	0.133	
γ (deg)	−5.0	0.5	
p (rad/s)	−6.61	0.661	
q (rad/s)	0.827	0.0827	
r (rad/s)	−2.58	0.258	

Spin is a hazardous phenomenon in which an aircraft has lost lift at a high α, and it causes fatal injuries, even with skilled pilot. Once α exceeds 21°, the aircraft stalls, and enters an un-commanded state of spinning (Martin, 1988). In general, the recovery technique from the state is well-known theoretically and empirically in various civil and military aircraft types (i.e., fighters with retreat angles or delta-type wings). Although how to recover is well known, early spin prognosis is crucial to give the pilot enough time to execute the optimal recovery control actions. Otherwise, spin would have been fully developed, and consequently diminishes the likelihood of recovery (Zhang & Mahadevan, 2020). Moreover, current pilots are not trained for spin, which takes multiple turns to recognize that the aircraft has entered a spin (DeLacerda, 2002). According to the report by the Civil Aviation Authority (CAA), there are three stages of spin, namely incipient, fully developed, and recovery, as shown in Fig. 5 (Bunge, 2017). Incipient spin is the stage from the moment the aircraft begins to stall and rotate to fully developed spin. Generally, spin usually takes one or two turns to enter the stage of unrecoverable spin, which is the main reason why this phase is essential for the development of recovery techniques and ML training models for spin prognosis. As the aircraft’s rotation rate and speed stabilize, it starts to fall vertically, leading to a fully developed spin. At this point, the aerodynamic forces and inertial forces are balanced; and hence, the angle of attack, as well as the rotational movement around the vertical axis, are constant. Finally, the final phase of spin is recovery, which is well established through pilot training. Although spin recovery is well known, early spin prognosis is crucial to give the pilot enough time to execute the optimal recovery control actions. To capture the initial signal of spin, detection and prediction of the incipient phase is essential in hindering the fully developed spin stage, at which turning state evolves (Bunge, 2017). As reported by AOPA (Bennett & Lawson, 2018), most spin accidents occurred at lower than 1,000 ft above the ground level elevation while it is recommended to allow a minimum of 2,000 ft above base height for abnormal attitude flight recovery to ensure there is enough room to complete the recovery process safely (Transport Canada, 2003; The Honourable Company of Air Pilots, 2022). Although the initial altitude of spin is almost traffic pattern altitude, the recovery is only possible when the altitude is sufficient (Civil Aviation Safety Authority (CASA), 2021). This proposed model predicts the altitude of flight that has the strongest correlation with spin, to avoid unrecoverable spin.

Figure 5 Typical occurrence of spin with three main phases, which are incipient, fully developed, and recovery phases.

Results and discussion

Neural network training and validation

This subsection thoroughly investigates the RNN-based model training and stability. As depicted in Fig. 6, both training and validation losses exhibit a monotonic decrease with training steps, indicating stability and convergence during training. This monotonic decrease, particularly in the validation loss, suggests that the model generalizes well and does not exhibit signs of overfitting. The training loss is slightly higher than the validation loss in the ML configuration due to the regularization effects of batch normalization and layer dropouts. These techniques help prevent overfitting but can lead to a slightly higher training loss. Layer dropouts, operating exclusively during ML training, further contribute to this difference. Additionally, the RMSE for the ML configuration, calculated across all five folds, averages 0.248 ft with a standard deviation of 0.0207 ft. The consistent performance across all k-folds further confirms the robustness of the ML configuration.

Figure 6 Loss function with training and validation data for the RNN-based model.

Upon ML training, the RNN-based model for predicting aircraft trajectories during spin upset conditions was evaluated. Altitude, a critical safety metric in air traffic management (ATM), exhibits significant loss during spin upset condition. Model predictions were compared against ground test data samples obtained from GTM simulations. The spin conditions for representative test data samples are given in Table 3. As shown in Fig. 7A, ML predicted altitude decreases linearly and closely aligns with GTM predictions across the simulated trajectory time. The absolute error percentage for the whole flight simulation time is given in Fig. 7B. In practical settings, recognizing the incipient phase of the spin upset, which lasts for only 4 to 6 s, is crucial for successful recovery. The absolute error of the ML model predictions increases exponentially with simulation time for the investigated cases, reaching up to 3.5%. However, during the incipient spin, the absolute error increases at a slowing rate, where the maximum error after 6 s is less than 0.5%. This shows that the model is robust in detecting the spin trajectory during its early phases.

Table 3 Test spin condition samples and their performance.

Spin conditions	Test sample 1	Test sample 2	Test sample 3	
eas (knots)	75.227	75.105	81.486	
tas (knots)	79.826	79.696	86.468	
α (deg)	21.286	20.392	21.375	
β (deg)	−0.13258	−4.3243	−0.78369	
γ (deg)	−4.4759	−4.7109	−5.1024	
p (rad/s)	−7.0722	−4.2907	−4.4548	
q (rad/s)	0.61657	0.87233	0.60827	
r (rad/s)	−2.7895	−1.6982	−1.7224	
Altitude error at 6 s of spin	0.43	0.15	0.09	

Figure 7 Sample comparison between the ML model and GTM predictions for (A) altitude during incipient and fully developed spin phases, (B) expected range of altitude absolute error.

Figure 8A shows the frequency distribution of altitudes for both the ground truth test data and the ML predictions throughout the total simulation time. The histogram demonstrates that the ML model captures the overall distribution of the ground truth data, although minor deviations are noticeable in specific altitude ranges. Figure 8B presents boxplots summarizing the central tendency and variability of the altitudes. Both datasets exhibit similar medians and ranges of altitude, suggesting that the ML model closely mirrors the ground truth data.

Figure 8 Comparison of altitude distributions: (A) histogram and (B) boxplot of ground truth and ML model for altitude prediction.

To further assess the model prediction accuracy, the model was tested on 100 new spin trajectories that were not used in the training and validation errors computation. Figure 9A illustrates the comparison between the ML model predictions and the ground truth mean altitude of the incipient spin phase over time, along with a 95% confidence interval. The close alignment between the predicted means and the confidence intervals provides further evidence supporting the accuracy of the proposed model, reinforcing the evaluation in Fig. 8. In addition, Fig. 9B presents a parity plot comparing the predicted incipient spin altitudes with the ground truth values. The plot includes a regression line with a coefficient of determination (R2) equals to 0.997, indicating a strong correlation between the predicted and actual values. This serves as an additional validation of the performance of ML model’s performance. However, one trajectory prediction exhibits a significant offset, which is likely an outlier. Despite this discrepancy, the overall results, including the high R2 value and consistent confidence intervals, suggest that the model provides accurate and reliable altitude predictions.

Figure 9 (A) Mean incipient spin altitudes of the ML model and ground truth, with 95% confidence intervals; (B) parity plot of predicted vs ground truth incipient spin altitudes for all 100 new test cases.

Model impact and challenges

As discussed in the Methodology section, spin has three temporal phases of incipient, fully developed, and recovery. To have a chance to recover, pilots should recognize the first sign of spin, which is the incipient spin phase. To solve this problem, this study provides the prediction of the first phase of the spin scenario for practical reasons, since there is no point in full trajectory prediction. In ATM, altitude loss is the one of the most critical parameters in flight trajectory planning and safety measurements, therefore, in this study, the in-flight upset prognosis method focuses on altitude prediction. Moreover, spin usually takes place at relatively low altitude, compared to other upset scenarios, hence spin is more critical and challenging for recovery. As shown in Fig. 10, the first two main phases of spin upset are captured through GTM simulations. In practice, the most critical period of spin is up to 6 s, where mid-air collision and accidents occur. Further, in the case of the spin recovery algorithm, the aircraft should be able to recover after 6 s to avoid further altitude loss and ultimate ground collision. Hence, the first 6 s of the spin scenario is selected as the region of interest for the ML model evaluation and is showing good agreement with the GTM predictions. Physics-based constraints, as well as further ML configuration iterations, can ultimately enhance the predictions accuracy as well as allow the modularity of model implementation into various aerodynamic upset scenarios (Huang, Wang & Wang, 2022).

Figure 10 Spin upset scenario phases prediction using GTM.

The ML model exhibits superior computational performance compared to the GTM. While the GTM requires approximately 3.3 min for predicting the overall spin upset trajectory, the ML model accomplishes this in just 1.66 s—a remarkable 99.2% improvement in computational efficiency. This enhancement positions the ML model as a powerful tool for next-generation ATM, enabling real-time predictions of complex nonlinear systems.

Despite its advantages, the RNN model encounters challenges such as vanishing gradients. Addressing this, the model incorporates additional hidden layers and batch normalization, improving prediction accuracy and training stability. However, for multi-variable complex predictions, LSTM and gated recurrent unit (GRU) NNs, which are variations of RNNs, have been designed to address such challenges through introducing dynamic memory, which determines the percentage of history variable to keep, and percentage to forget (Hamza et al., 2023). The effect of dynamic memory allocation and NN tuning on full aerodynamics response prediction accuracy will be further studied as future work. Additionally, the developed RNN-based model is purely data-driven, where the physics of the real-world system are captured through minimizing the RMSE of the predicted and ground truth data. However, physics-based NN architectures can be implemented to ensure that the nonlinear six-degree-of-freedom aerodynamic relations are not being violated in trajectory predictions. This can be achieved through penalty regularization terms, as well as formulating the ML model loss function as a reduced objective function which generates a multi-dimensional design space accounting for the non-linear aerodynamic constraints (Hamza et al., 2023). Finally, fusing aerodynamic constraints within the RNN-based model will allow current model extension to simulate various flight upset conditions.

Conclusions

This study is highly relevant to aviation safety, particularly in addressing flight spin upset conditions. By integrating a high-fidelity simulation environment with NN architectures, the proposed RNN-based model offers a solution for real-time spin prognosis. Designed as a machine learning tool, the model accurately mimics complex nonlinear aircraft trajectory behavior, handling the temporal flow of time-series data from takeoff to landing. It is particularly effective in predicting critical flight parameters during the onset of spin, ensuring computational efficiency for integration into ATM automated tools.

The RNN-based model leverages hidden layers with batch normalization to ensure stability, while scaling through standardization allows smooth convergence despite the diverse scale of input parameters. RMSE is applied as the loss function to capture the difference between predicted and ground truth spin trajectories. Minimizing RMSE for GTM ground truth data ensures realistic correlations among equivalent airspeed, angle of attack, sideslip angle, roll, pitch, yaw rates, and spin altitude. The model demonstrates remarkable computational efficiency, improving post-spin upset prediction speed by 99.2% compared to high-fidelity reference simulations, while maintaining accurate predictions during the critical early stages of spin occurrence.

Integration the RNN-based model into aircraft systems can support pilots by reducing their cognitive load and aiding in rapid response during critical spin situations, especially at low altitudes. By enabling early anomaly detection, the model shows potential for reducing fatality rates in high-risk spin scenarios, serving as a practical and robust tool for aviation safety enhancement. While the model has demonstrated significant potential, certain limitations remain. Primarily, the model is optimized for the initial stages of spin detection, and its accuracy declines as the spin progresses, indicating limitations in predicting the entire spin sequences. Additionally, this model currently designed specifically for spin scenarios, which restricts its application to other types of flight anomalies. The reliance on simulated data for training also presents a limitation, as real-flight conditions introduce variables such as weather changes and aircraft-specific characteristics that may impact model performance. Expanding the training dataset with real-flight data could improve model generalizability in future applications.

The next phase of this research will focus on expanding the model’s scope beyond spin upset scenarios to encompass general flight trajectory prediction in real time. Case studies on other flight anomalies such as sudden altitude loss, control loss in various states will be conducted to broaden the model’s applicability. Additionally, the use of more complex NN models could enhance the model’s ability to predict complex trajectory behaviors across all stages of flight. Finally, incorporating physics constraints in the RNN architecture could further improve predictive reliability, adapting the model to a wide range of flight conditions and enhancing its role in aviation safety technology.

Supplemental Information

Supplemental Information 1 Data-Driven RNN Model Development Data Archive.

Supplemental Information 2 Python Code for Data-Driven RNN Model Development.

Additional Information and Declarations

Competing Interests

The authors declare that they have no competing interests.

Author Contributions

Naeun Kim conceived and designed the experiments, performed the experiments, analyzed the data, performed the computation work, prepared figures and/or tables, authored or reviewed drafts of the article, and approved the final draft.

Mohamed H. Hamza conceived and designed the experiments, performed the experiments, analyzed the data, performed the computation work, authored or reviewed drafts of the article, and approved the final draft.

Bong-Hwan Koh conceived and designed the experiments, authored or reviewed drafts of the article, and approved the final draft.

Data Availability

The following information was supplied regarding data availability:

Code and raw data are available in the Supplemental Files.

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
