# Peer review of "Data-driven flight path monitoring technique using recurrent neural network for the safety management of commercial aircraft"

_PeerJ Computer Science, doi:10.7717/peerj-cs.2753_

## Round 0.1 · original submission · Major Revisions

Based on the reviewer comments, the manuscript must be revised. The reviewer suggested some referentes, but its suitability must be evaluated.

Reviewer 1 ·

Basic reporting

As below

Experimental design

The description of the RNN-based model's development lacks technical detail. Readers would benefit from a more comprehensive explanation of the model architecture, training process, and validation procedures to understand its efficacy in predicting flight trajectories accurately.

While the manuscript claims that the trained RNN model showed accurate results compared to GTM, it lacks quantitative data and statistical analysis to support this claim. The absence of specific performance metrics and comparative results undermines the credibility of the findings.

Validity of the findings

The conclusion should offer insights into the practical implications of the study's findings for improving air traffic management and aviation safety. Additionally, it should highlight any limitations or future research directions to provide a well-rounded evaluation of the study.

Overall, the manuscript presents an important topic but falls short in providing sufficient detail, justification, and empirical evidence to support its claims. A more rigorous and comprehensive approach to research methodology, data analysis, and result interpretation is necessary to enhance the credibility and impact of the study in the field of aviation safety.

Additional comments

The manuscript entitled “Data-driven flight path monitoring technique using recurrent neural network for the safety management of commercial aircraft” has been investigated in detail. The manuscript addresses the critical issue of spin/stall upset scenarios in aviation safety and proposes an RNN-based model as a surrogate for predicting flight trajectories in real-time. However, it lacks clarity in problem statement articulation, methodology justification, and empirical evidence presentation. A more detailed explanation of the problem's significance, the rationale behind the chosen methodology, and quantitative validation results is needed to strengthen the study's impact and credibility. There are some points that need further clarification and improvement:
1) The introduction fails to provide a clear and concise overview of the problem statement and the significance of addressing spin/stall upset scenarios in aviation safety. It lacks a thorough explanation of why human error in air traffic management, particularly in spin/stall situations, is a critical concern.
2) The rationale for the choice of methodology, specifically using a Recurrent Neural Network (RNN) as a surrogate for the Generic Transport Model (GTM) of NASA, is not sufficiently justified. The manuscript should provide a more detailed explanation of why an RNN-based model was selected and how it addresses the limitations of GTM.
3) The description of the RNN-based model's development lacks technical detail. Readers would benefit from a more comprehensive explanation of the model architecture, training process, and validation procedures to understand its efficacy in predicting flight trajectories accurately.
4) While the manuscript claims that the trained RNN model showed accurate results compared to GTM, it lacks quantitative data and statistical analysis to support this claim. The absence of specific performance metrics and comparative results undermines the credibility of the findings.
5) The conclusion should offer insights into the practical implications of the study's findings for improving air traffic management and aviation safety. Additionally, it should highlight any limitations or future research directions to provide a well-rounded evaluation of the study.

Overall, the manuscript presents an important topic but falls short in providing sufficient detail, justification, and empirical evidence to support its claims. A more rigorous and comprehensive approach to research methodology, data analysis, and result interpretation is necessary to enhance the credibility and impact of the study in the field of aviation safety.

Reviewer 2 ·

Basic reporting

The manuscript titled "Data-driven ûight path monitoring technique using recurrent neural network for the safety management of commercial aircraft". Authors developed a Recurrent Neural Network (RNN)-based model as a surrogate to mimic GTM simulations and predict nonlinear ûight response under spin/stall upset with computational eûciency feasible for real-time application. The trained RNN model showed accurate results compared to GTM in predicting the ûrst few seconds of the aircraft trajectory under spin/stall condition, which is vital for spin/stall prognosis and immediate recovery.


Generally speaking, the paper weak, written badly, but it can be improved. however the following major corrections are suggested.

Experimental design

Materials & Methods section is very very badly written. There should be a research Framework / architecture (in the form of diagram) with explanation. Sample dataset should also be included in Materials & Methods section.

Validity of the findings

1. Results are not clearly visible in Results section. To increase the visibility of your work done, The trained model results should be shown in tables with graphical representations. (A brief debate on results and graphs are needed as well).

2. Results section and Discussion section should be combined. Authors have included them separately. Rename this section to "Results and Discussions"

Additional comments

1. After Abstract KEYWORDS are missing.

2. Avoid using word "We" and "Our" in the manuscript.

3. There should be a separate detailed section of Literature Review (LR) / Related work after Introduction section, which is missing. That is not good practice and that should be taken care of. At least 10 to 12 research articles of last 5 years have to be summarized with references in this section, this will help you to justify gap / Your contribution.

4. The problem statement should be provided precisely.

5. There should be a brief debate regarding your contribution in Introduction section.

6. Machine learning, Neural Network and other used techniques must be explained (in separate section) with references in your paper. This section should be incorporated after Related work section.

7. The punctuation and grammar of the manuscript should be improved considerably.

Reviewer 3 ·

Basic reporting

I have read the entire manuscript and found it interesting and in optimal form. I do not recommend any further changes.

Experimental design

no comment

Validity of the findings

no comment

---

## Round 0.2 · Minor Revisions

Please address the comments of reviewer 2

Reviewer 1 ·

Basic reporting

My comments have been addressed. It is acceptable in the present form.

Experimental design

My comments have been addressed. It is acceptable in the present form.

Validity of the findings

My comments have been addressed. It is acceptable in the present form.

Reviewer 2 ·

Basic reporting

Authors have shown Research framework of methodology section in response document. but this framework is not present in paper.

same with results and discussion. results are not visible in paper.

rest is ok

Experimental design

same with results and discussion. results are not visible in paper.

Validity of the findings

No

Additional comments

No

---

## Round 0.3 · accepted · Accept

The manuscript was accurately revised, and it is ready for publication.